

# Proportional feature pyramid network based on weight fusion for lane detection

Jiapeng Hui[1,2], Guoyun Lian[1], Jiansheng Wu[2], Shuting Ge[1,2] and Jinfeng Yang[1]

[1] Institute of Applied Artificial Intelligence of the Guangdong-Hong Kong-Macao Greater Bay Area, Shenzhen Polytechnic University, Shenzhen, Guangdong, China
[2] School of Computer and Software Engineering, University of Science and Technology Liaoning, Anshan, Liaoning, China

## ABSTRACT

Lane detection under extreme conditions presents a highly challenging task that requires capturing each crucial pixel to predict the complex topology of lane lines and differentiate the various lane types. Existing methods predominantly rely on deep feature extraction networks with substantial parameters or the fusion of multiple prediction modules, resulting in large model sizes, embedding difficulties, and slow detection speeds. This article proposes a Proportional Feature Pyramid Network (P-FPN) through fusing the weights into the FPN for lane detection. For obtaining a more accurately detecting result, the cross refinement block is introduced in the P-FPN network. The cross refinement block takes the feature maps and anchors as inputs and gradually refines the anchors from high to low level feature maps. In our method, the high-level features are explored to predict lanes coarsely while local-detailed features are leveraged to improve localization accuracy. Extensive experiments on two widely used lane detection datasets, The Chinese Urban Scene Benchmark for Lane Detection (CULane) and the TuSimple Lane Detection Challenge (TuSimple) datasets, demonstrate that the proposed method achieves competitive results compared with several state-of-the-art approaches.

## INTRODUCTION

Lane detection has received widespread attention as an essential component of advanced driver assistance systems and autonomous driving technologies (*Badue et al., 2021*). Accurate and efficient lane detection provides crucial information for autonomous driving systems, such as lane departure marking, lane-keeping assistance, and adaptive cruise control (*Zhang et al., 2021*). As a fundamental aspect of vehicle perception, many researchers have devoted their efforts to developing efficient and accurate lane detection algorithms to achieve reliability and practicality in various environments. However, lane detection still encounter some challenges for detecting accurate lanes, such as illumination variations, severe weather condition, difference lane marking, and vehicle moving directions *etc*.

Corresponding author
Guoyun Lian,
lianguoyun@szpu.edu.cn

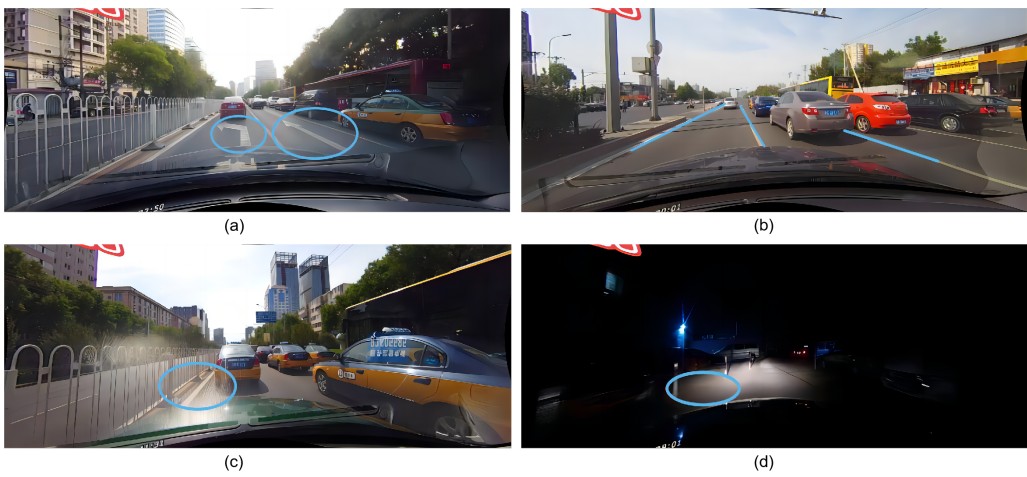

**Figure 1** **(A) The landmark and lane line have the similar characteristics, but it difficult to distinguish them. (B) The lane lines were successfully detected, but the positions are not accurately localized. (C, D) It is challenging to detect lane in severe situations without the fusion of global contextual feature information.** Image source: CULane dataset, https://xingangpan.github.io/projects/CULane.html.

The landmark is a particular geometric structure with rich semantic information. The lane line and landmark share the similar feature, but they have different semantics. Only through the feature, it is difficult to distinguish them. The high-level semantic information and low-level feature are complementary for accurate lane detection. However, the information fusion from different levels remains an unsolved problem. From Fig. 1A, it can be observed that the landmark and lane line have the similar characteristics, but it difficult to distinguish them. Figure 1B shows that the lane lines were successfully detected, but the positions are not accurately localized. Figures 1C and 1D demonstrate that without the fusion of global contextual feature information, it is challenging to detect lane in severe situations. Therefore, the semantic information is very important for lane detection.

Recently, the lane detection methods based on information fusion were proposed (*Pan et al., 2018*). For example, the standard Feature Pyramid Network (FPN) was used to tackle the global contextual information fusion and the lane detection method based on this network was proposed (*Wang et al., 2022*). However, these approaches directly integrate feature maps from different levels and input them into subsequent detection modules without considering the different contribution of different level feature map, which is very important for different level to detect the lane.

In this article, a Proportional Feature Pyramid Network (P-FPN) is proposed for lane detection, in which the low-level and high-level features are employed and the feature layer with more valuable information will be assigned greater weight. Specifically, the high semantic features is firstly performed detection to coarsely localize lanes. Then, the refinement module is conducted based on weight-fusion strategy to get more precise locations. Our model focuses on capturing more global contextual information by learning from informative feature layers. The main contributions of this article can be summarized as follows:

- A novel Proportional Feature Pyramid Network (P-FPN) is proposed for lane detection, in which the low-level and high-level features are fully utilized.
- The weight-fusion algorithm has been proposed to improve the performance and robustness of the lane detection model by adapting to the different data distributions.
- Experiments were conducted on the widely used datasets, CULane (*Pan et al., 2018*) and TuSimple dataset, achieving higher detection accuracy and competitive detection speed.

The rest of the article is organized as follows. The literature review section discusses the current related works of the lane detection and its challenges. The methodology section introduces the overall architecture of the proposed framework, including Proportional Feature Pyramid Network (P-FPN), cross refinement block and training strategy. The experimental results section presents the lane detection performance of the proposed method. Finally, the conclusion section summarizes the whole article.

## LITERATURE REVIEW

Accurate and efficient lane detection plays a crucial role in optimizing traffic decisions (*Subotić et al., 2022*), and its synergistic interaction with road marking recognition provides essential support for intelligent driving (*Jayapal, Muvva & Desanamukula, 2023*). At present, lane detection methods can be classified into two classes including traditional computer vision approaches (*Berriel et al., 2017*; *Assidiq et al., 2008*) and deep learning methods (*Hou et al., 2019*; *Philion, 2019*).

Traditional lane detection methods commonly utilize the image processing techniques, such as edge detection (*Zhou et al., 2010*; *Yoo et al., 2017*), Hough transform (*Liu, Wörgötter & Markelić, 2010*), *etc*. The main advantage of these methods is their computing speed, but in complex urban driving scenes or varying lighting or poor weather conditions (*Sun, Tsai & Chan, 2006*; *Bar Hillel et al., 2014*), the performance of these methods will be severely degraded.

Benefiting from the effective feature representation of Convolutional Neural Networks (CNN), many lane detection methods (*Hou et al., 2019*; *Roberts et al., 2018*; *Zou et al., 2020*) have achieved excellent performance. In these methods, parameter-based lane detection typically employ curve fitting techniques to model the lane lines as polynomial curves. Other works (*Gansbeke et al., 2019*; *Feng et al., 2022*) proposed fit polynomial curves to deep neural networks. These approaches can handle the lane lines with different shapes and curvatures, and aid to address the occlusion issues. But these methods are more sensitive to variations of the lane lines.

The methods based on semantic segmentation, such as MobileNet (*Howard et al., 2017*), ERFNet (*Romera et al., 2018*), LaneNet (*Wang, Ren & Qiu, 2018*), *etc*., have been proposed for lane detection. In these methods, the pixels of the image were classified into lane or background. The advantage of these approaches is able to effectively detect the lane in complex scenes and diverse lighting conditions. Since it is easy to misposition the lane line for the segmentation methods, the cumbersome post-processing is needed. Also, when facing occlusion or bright illumination, the performance of the segmentation method will be degraded.

Currently, the anchor-based lane detection is one of the main lane detection methods. Anchor-based methods typically adopt the object detection techniques to detect the positions and shapes of lane lines by setting the anchors in the image, in which the lane detection is transformed into a classification problem. The advantage of the methods is able to handle the lane lines with different shapes. The anchor-based approach with a transformer model (*Liu et al., 2021b*) was proposed and achieved high accuracy in lane detection. *Tabelini et al. (2021a)* introduces a novel anchor-based attention mechanism that utilizes global information to accurately determine the position of lane markings. Similar to *Tabelini et al. (2021a)*, *Qin, Wang & Li (2020)* demonstrates that the reduction of the anchor size can improve the detection speed. *Zheng et al. (2022)* explores a feature aggregation module that iteratively enhances anchoring within feature maps across several levels, resulting in favorable results. However, it is difficult to find the start points of the lane lines for the anchor-based methods in some complex situations, which will result in inferior performance.

Inspired by the Feature Pyramid Network (FPN) (*Lin et al., 2017a*), it is a architecture that has been specifically developed to address the challenge of integrating multi-scale information. The FPN has demonstrated significant advancements in the fields of object detection and semantic segmentation. The FPN model integrates bottom-up and top-down feature maps by employing a horizontal connection mechanism. Specifically, the technique of horizontal connecting splices involves combining feature maps from several levels in order to create a feature pyramid that possesses both high resolution and abundant semantic information. The utilization of a pyramid structure in FPN facilitates the detection and segmentation of objects across various scales uniformly. *Zheng et al. (2022)* explored to integrate the FPN and anchors to detect the lane and obtained good performance.

However, these models employ predetermined learning strategies for feature maps at different levels and allocate the same attention to the feature layers with different information, which resulting in the acquisition of extraneous features. Moreover, these methods based on the anchors suffer from over-dependence on the specific dataset.

## METHODOLOGY

The overall architecture of the proposed framework based on weight fusion and cross refinement for lane detection is illustrate in Fig. 2. It primarily consists of two sequential components: the Proportional Feature Pyramid Network(P-FPN) block and the Cross Refinement block.

### Proportional Feature Pyramid Network (P-FPN)

Since lane lines usually occupy a small proportion of an image, lane detection can be regarded as the small object detection. Therefore, each individual pixel belongs to the lane line is very important for detection, and even a small number of pixels can significantly affect the final detection result.

In reference to *Zheng et al. (2022)*, a lane line can be represented as a sequence of points along the $y$-axis, with the fixed pixel intervals for sampling the $x$-axis. This representation can be denoted as $L = \{(x_1, y_1), (x_2, y_2), \ldots, (x_n, y_n)\}$. An anchor for lane lines consists

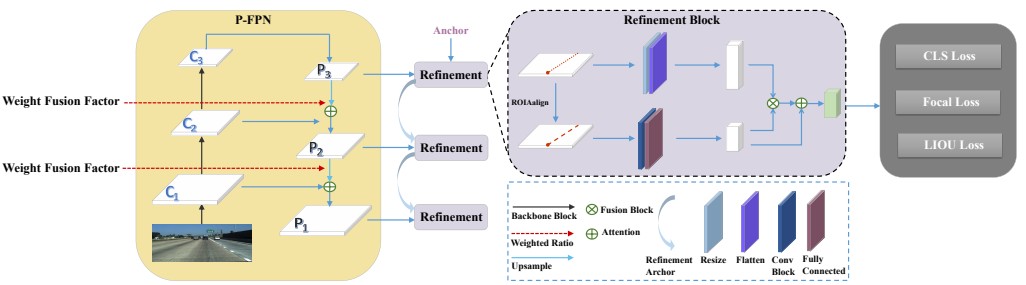

**Figure 2 Overview of the proposed method.** It consists of P-FPN and Refinement Block. The feature maps ($C_i$) are acquired using ResNet (*He et al., 2016*), subsequently fed into the P-FPN. Then, the weight fusion factor is employed to focus on the feature layers with more context information. Finally, the fusion feature maps ($P_i$) are fed into the refinement block module to refine the lane lines.

of four components: (1) foreground and background probabilities;(2) the length of the anchor along the $y$-axis; (3) the start position of the anchor and its angle $\theta$ concerning the $x$-axis; (4) the offset of n points relative to the anchor line.

In the proportional feature pyramid network (P-FPN), the traditional feature extraction networks, ResNet (*He et al., 2016*), is employed to extract the semantic feature. The model uses the last three layers of the backbone as the original input feature maps. These feature maps are then progressively fused at different levels to be used by subsequent feature refinement modules. Then the weight fusion between high-level and low-level semantic features is explored based on the relative proportion of each level feature. Therefore, our model can provide greater attention to the feature layers with more context information for each image.

In FPN-based lane detection methods, the performance was primarily influenced by two factors: the downsampling factor and the weight fusion mechanism employed between adjacent layers.

Previous works (*Ren et al., 2017*; *Lin et al., 2017b*; *Kong et al., 2020*; *Tan, Pang & Le, 2020*) have extensively investigated the downsampling factor and concluded that the lower downsampling factors lead to better performance, albeit with huge computational cost.

The weight fusion mechanism between adjacent feature layers in P-FPN is shown in Eq. (1).

$$P_i = f_{Conv}(C_i) + W_{i+1} \times f_{UpSample}(P_{i+1}) \tag{1}$$

where $f_{Conv}(\cdot)$ represents a 1×1 convolution operation that is employed to ensure consistency in the channel dimensions, $f_{Upsample}(\cdot)$ executes a 2 × upsampling on the higher-level feature maps in order to align the resolution, $W_{i+1}$ denotes the weight fusion factor between adjacent feature layers, $C_i$ represents the $i$th level feature map and $P_i$ denotes the $i$th level fusion feature map in the P-FPN network. The purpose of P-FPN is to add a suitable weight fusion factor $W$ to the feature fusion process.

To further explore how to get the effective weight fusion factor $W$, a weight fusion algorithm is designed, which can improve the accuracy of the lane detection through

optimizing the weight assigned to each data point. At different levels of the P-FPN network, the data distribution on the feature map is different. In order to train the different weight features for each data point, the weight fusion algorithm is proposed as shown in Algorithm 1, which is outlined in the following section.

---

**Algorithm 1** Algorithm 1

---

**Require:** $N$ (List of all image indices)
**Require:** $List_{Ln}$ (list of tuples containing image index and three LIOU losses $L_{p1}$, $L_{p2}$, $L_{p3}$)
**Require:** $\alpha$ (hyperparameter value)
**Ensure:** $List_{Wn}$ (list of tuples containing image index and two weight fusion factors $W_3^2$ and $W_2^1$)

  1: **function** MATCHLOSS                     ▷ Retrieve losses for image indices
  2: **end function**
  3: **function** CALWEINUM                ▷ Compute weight fusion factors
  4: **end function**
  5: $List_{Ln} \leftarrow []$
  6: **for** $Ni$ in $N$ **do**
  7:     $List_{Ln}$. append$((N_i, L_{ip1}, L_{ip2}, L_{ip3}))$
  8: **end for**
  9: $List_{Wn} \leftarrow []$
10: **for** $(N_i, L_{ip1}, L_{ip2}, L_{ip3})$ in $List_{Ln}$ **do**
11:     $(W_3^2, W_2^1) \leftarrow$ CALWEINUM$(List_{Ln}, \alpha)$
12:     $List_{Wn}$. append$((N_i, W_3^2, W_2^1))$
13: **end for**
14: **return** $List_{Wn}$

---

The algorithm proceeds as follows: (1) The initial FPN is employed for training the model with a weight fusion factor of The *LIoULoss* values (*Zheng et al., 2022*) of all feature layers will be recorded; (2) the weight of each layer in the P-FPN network is calculated through the Eq. (2); (3) the weight fusion factor between adjacent layers is calculated using Eq. (3); (4) save the weight fusion factors and the index of the input image.

$$S_i = e^{-\alpha \times \mathcal{L}_{LIoU}(i)} \tag{2}$$

where the variable $e$ denotes Euler's number, which is a constant representing the base of the natural logarithm, $\mathcal{L}_{LIoU}(\cdot)$ denotes the line intersection over union loss (*Zheng et al., 2022*), $\alpha$ is a hyperparameter which can be set as from 0.1 to 0.9, its optimal value in different datasets is selected through experimenting which is elaborated in our experiment section. The weight fusion factor can be calculated as follows:

$$W_i = \frac{S_i}{S_{i-1}} \tag{3}$$

where $W_i$ is the weight ratio of the $i$ and $i-1$ layer, $S_i$ represents the weight of the $i$th layer. $i$ can be set as 3 or 2.

## Cross refinement block

The lane detection with high-level features from the P-FPN can be localize lanes coarsely. Then, the coarsely detected lanes can be further refined by using the cross refinement block, which takes the anchors and the fused feature maps from the P-FPN as the input of the block.

The anchors refine using ROIAlign to obtain more precise features (*He et al., 2020*). The delicate anchor features aggregate through convolutional layers. These features are further processed using attention-weighted operations with feature maps that have changed dimensions and sizes. Therefore, the feature maps are resized and flattened to adjust their size and dimensions, while the anchors undergo fine-grained operations through ROIAlign. After that, convolutional and fully connected operations are applied to align the dimensions and shape with the transformed feature maps. The anchors initially map the feature maps to obtain an attention matrix $M$. The formula is shown in Eq. (4).

$$M = \frac{X_p^T X_f}{\sqrt{C}} \tag{4}$$

where $X_p$ represents the anchor's representation, $X_f$ represents the global feature map information, the aggregation matrix $W$, which refines the feature map's aggregation over anchors, is obtained through a weighted process. The formula is shown in Eq. (5).

$$W = M X_f \tag{5}$$

Finally, the output is added to the anchor $X_p$.

## Model training
### *Positive sample allocation*

During the training process, the selection of the top$-$k predictions is performed for each target through sorting all anchors using the following cost function. The sum of the *LIoU* values for the k predictions is calculated and rounded to yield the positive predictions for the target, which are referred to as $K_{pos}$. The cost function is subsequently employed to designate the $K_{pos}$ predictions with the minimum cost as positive samples. The cost function is represented by Eq. (6).

$$C_{price} = W_{sim} C_{sim} + W_{cls} C_{cls} \tag{6}$$

$$C_{sim} = (C_{dis} C_{xy} C_{theta})^2 \tag{7}$$

where $C_{cls}$ denotes the focus distance, a metric that measures the separation between the anticipated trajectory and the actual path at their respective places of focus. The similarity cost $C_{sim}$ represents the cost associated with the similarity of the three criteria $C_{dis}$, $C_{xy}$, and $C_{theta}$. The variable $C_{dis}$ denotes the mean pixel distance separating valid lane points, $C_{xy}$ signifies the distance between the initial points, and $C_{theta}$ quantifies the horizontal angle difference. The values have been adjusted to fall inside the range of [0, 1]. Furthermore, the predetermined weight fusion factors $W_{sim}$ and $W_{cls}$ are employed to modify the relative contributions of each standard.

*Training Loss*

As discussed above, the lane can be represented as discrete points which needed to be regressed with the ground truth. The distance loss is often used to regress these points, which will result in less accurate regression (*Zheng et al., 2022*). In reference to *Zheng et al. (2022)*, the Line Intersection over Union (LIoU) loss $\mathcal{L}_{LIoU}$, which is the ratio of intersection over union between two line segments, is defined as:

$$\mathcal{L}_{LIoU} = 1 - LIoU \tag{8}$$

where $LIoU$ can be computed as

$$LIoU = \frac{\sum_{i=1}^{N} d_i^{\vartheta}}{\sum_{i=1}^{N} d_i^{u}} \tag{9}$$

where $d_i^{\vartheta}$ represents the intersection between the predicted and labeled lines, $d_i^{u}$ represents the union of the predicted and labeled lines. In the context of optimizing non-overlapping line segments, it is possible for the value of $d_i^{\vartheta}$ to be negative, so the value of LIoU is within the range of $(-1, 1)$.

This study employs three different loss functions to collectively supervise the training process. $\mathcal{L}_{cls}$ represents the classification loss, $\mathcal{L}_{xytl}$ corresponds to the regression loss for the starting point position, angle, and length, and $\mathcal{L}_{LIoU}$, as proposed in *Zheng et al. (2022)*, performs regression on the lane lines as a whole unit.

$$\mathcal{L}_{total} = w_{cls}\mathcal{L}_{cls} + w_{LIoU}\mathcal{L}_{LIoU} + w_{xy}\mathcal{L}_{xy} \tag{10}$$

During the training, the model will output the detection result of each layer through the cost function and input it into the loss function optimization model, and only output the detection result of the last layer during the test.

# EVALUATION METRICS

In order to compare the performance to other competitors in TuSimple and CULane, there are two evaluation metrics, accuracy and F1 score, often used for lane detection. The accuracy is used as the official evaluation metrics of TuSimple and CULane proposed in the literature. In reference to *Pan et al. (2018)*, The F1 score is adopted as the metric in our exprement, which is a value that represents a harmonic mean, and it is considered a reliable metric that combines precision and recall. They are calculated as follows:

## Accuracy

Accuracy is defined as the ratio of the number of correctly predicted instances to the total number of samples. This can be represented by the formula as shown in Eq. (11).

$$Accuracy = \frac{T_p + T_n}{T_p + T_n + F_p + F_n} \tag{11}$$

where $T_p$, $T_n$, $F_p$, and $F_n$ denote true positive, true negative, false negative and false positive rates, respectively.

## F1 Score

The F1 score is the harmonic mean of precision and recall. Precision measures how many of the positive predictions made by a model are correct, while recall measures how many of the positive examples in the data are correctly identified by the model. The F1 score combines these two metrics into a single number, which provides a balance between precision and recall, and is particularly suitable when there is an imbalance in the distribution of class labels. The formula for calculating the F1 score is shown as Eq. (12).

$$F1 = \frac{2 \times Precision \times Recall}{Precision + Recall} \tag{12}$$

where *recall* is defined as the ratio of correctly classified values to the total number of values in the dataset, *precision* is provided as the ratio of correctly predicted values to the total number of predicted values. The *recall* and *precision* are calculated as Eq. (13) and Eq. (14), respectively.

$$Recall = \frac{T_p}{T_p + F_n} \tag{13}$$

$$Precision = \frac{T_p}{T_p + F_p} \tag{14}$$

where $T_p$, $F_p$, and $F_n$ denote true positive, false negative and false positive rates, respectively.

Similar to *Zheng et al. (2022)*, in our experiments, the accuracy and F1 score are all employed for the Tusimple dataset. The F1 score is employed as the evaluation metric for the CULane dataset.

# EXPERIMENTAL RESULTS

## Dataset and setting

### TuSimple

The TuSimple dataset is one of the widely used datasets in lane detection, consisting of 6,408 road images on US highways, of which the images are under different traffic and weather conditions. The dataset contains highway scene with 3,268 images for training, 2,782 for testing, and 358 for validation. All the images have 1,280 × 720 pixels. In this dataset, in reference to *Liu et al. (2021b)*, the lane is treated as a 20-pixel-width line. If the predicted point is within the 20-pixel-width line, this point is considered as a correct point. If the number of correct predicted points of a lane is greater than 85%, the lane prediction is considered as a true positive (TP).

### CULane

The CULane dataset is a well-known, extensively used, and large scale challenging dataset for lane detection, initially presented in the SCNN (*Pan et al., 2018*). A total of 133,235 frames were extracted from more than 55 h of videos. The dataset contains nine challenging categories which are normal, crowded, highlighted, shadow, no line, arrow, curve, cross, and night respectively. The dataset consists of 88,880 samples for training, 9,675 for

**Table 1  Settings of the CULane and TuSimple datasets.**

| DataSet | CUlane | TuSimple |
|---|---|---|
| Train | 88,800 | 3,268 |
| Val | 9,765 | 358 |
| Test | 34,680 | 2,782 |

**Table 2  Proportion of challenging scenarios on the CULane test set.**

| Type | Percentage (%) |
|---|---|
| Normal | 27.7 |
| Corwded | 23.4 |
| Night | 20.3 |
| No line | 11.7 |
| Shadow | 2.7 |
| Arrow | 2.6 |
| Dazzle light | 1.4 |
| Curve | 1.2 |
| Crossroad | 9.0 |

validation, and 34,680 for testing. The testing data contains eight distinct challenging categories. All the images have $1,640 \times 590$ pixels. If the predicted point is within the 30-pixel-width of the ground truth, this point is considered as a correct point (*Zheng et al., 2021*). If the number of correct predicted points of a lane is greater than 50%, predicted lane is considered as a true positive (TP).

The details of the two datasets are shown in Table 1, and the proportion of each category in the CULane test set is shown in Table 2.

## Implement details

Our model is implemented based on PyTorch 1.8 and CUDA 11.2 on the Ubuntu 20.04 with RTX 3080 GPU to run all the experiments. All the input images are initially resized to $320 \times 800$ pixels. To augment the data for making the model robust indifferent lightness situations, similar to *Zheng et al. (2022)*, random affine transformation (translation, rotation, and scaling) (*Polson & Scott, 2011*) and random horizontal flips are employed in the experiments. Specifically, every input image is rotated a random degree within 10 degrees, scaled with a random factor between 0.8 and 1.2, and added a random brightness with the range $(-10, 10)$, which make the model adapt the same image in different situation. Especially in the evening, the model can minimize the influence of the low luminescence, and maintain good recognition quality. After preprocessing, the images are normalized to $320 \times 800$ pixels again.

In training phase, the quantity of anchors has been set to 72. The distribution cost is defined as $W_{cls} = 1$ and $W_{sim} = 3$. For the CULane dataset, the number of epochs was set to 30 with a batch size of 24. Vertical sampling was performed from 589 to 230 at intervals of 20 pixels. Corresponding loss weights were configured as $w_{cls} = 2.0$, $w_{LIoU} = 2.0$, and $w_{xy} = 0.2$. For the TuSimple dataset, the number of epochs was set to 70 with a batch

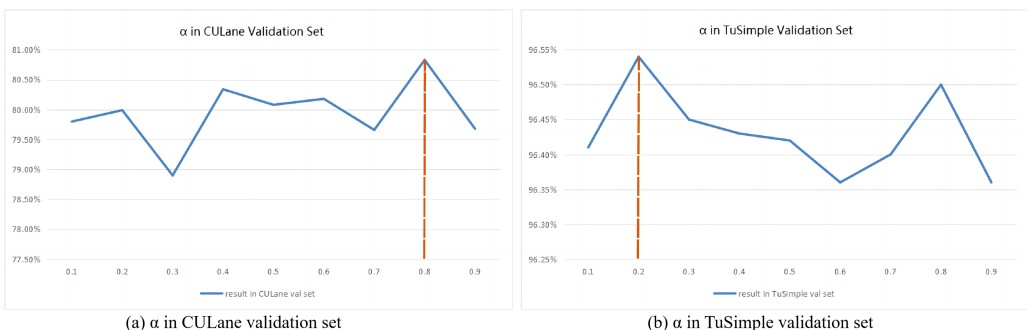

(a) α in CULane validation set    (b) α in TuSimple validation set

**Figure 3  Result of validation set on CULane and TuSimple dataset.** (A) The best $\alpha$ value of CULane validation set is 0.8, and (B) the best $\alpha$ value of TuSimple validation set is 0.2.

size of 40. Vertical sampling was performed from 710 to 150 at intervals of 10 pixels. Corresponding loss weights were configured as $w_{cls} = 6.0$, $w_{LIoU} = 2.0$, and $w_{xy} = 0.5$.

During the optimizing process, the AdamW (*Loshchilov & Hutter, 2019*) optimizer is used with an initial learning rate of 1e−3. The learning rate was decayed by a cosine annealing learning rate strategy with a decay factor of 0.9 (*Loshchilov & Hutter, 2017*).

## Analysis of hyperparameter $\alpha$ values

The weight calculation formula based on hyperparameters $\alpha$ is shown in Eq. (2). A higher $\alpha$ value has a larger penalty for model misdetecting, which makes the model be more suitable for learning complex samples. However, a lower $\alpha$ value has a smaller liability for false positives, which makes the model be easy to learn the simple data. The $\alpha$ values are different for different datasets. As shown in Fig. 3A, the experiments are conducted on the validation set of the CULane dataset, and it can be seen that the F1 score reaches a peak when the $\alpha$ value is 0. 8. On the TuSimple dataset, 20% of the training set is randomly divided as the validation set. As shown in Fig. 3B, it can be seen from the experimental results that when the $\alpha$ value is 0. 2, the F1 score reaches the peak. Therefore, in the following experiments, the $\alpha$ value is set to 0.8 for the CULane dataset and 0.2 for the TuSimple dataset.

## Ablation experiments of weight fusion mechanism

A series of ablation experiments are conducted to test the effectiveness of our proposed P-FPN network. In these experiments, three settings, layer $P3 \rightarrow P2$, $P2 \rightarrow P1$, and $P3 \rightarrow P2 \rightarrow P1$, are considered to perform refinement and the comparative experiments are designed between our proposed method (P-FPN) and the baseline method (CLRNet) (*Zheng et al., 2022*). The experimental results are shown in Table 3. It can be seen from the result, the detection performance of our method (P-FPN) surpasses the baseline method in all three settings, which proves the effectiveness of our weight fusion mechanism. Furthermore, adopting the refinements from $P3$ to $P2$ to $P1(P3 \rightarrow P2 \rightarrow P1)$ is much better than the other settings, which validates our final weight fusion mechanism can utilize low-level and high-level features better.

**Table 3** **Ablation studies were conducted on different stages of the pyramid network.** $P_i$ represents the $i$th fusion feature map in the network, and "with P-FPN" indicates the results using our proposed proportional feature pyramid network. Bold numbers are the best results.

| Settings | F1 in CULane (%) | F1 in TuSimple (%) | Accuracy in TuSimple (%) |
|---|---|---|---|
| P3 → P2 | 79.12 | 96.80 | 95.92 |
| P3 → P2 with P-FPN | 79.23 | 97.02 | 96.14 |
| P2 → P1 | 79.19 | 97.01 | 96.21 |
| P2 → P1 with P-FPN | 79.34 | 97.24 | 96.44 |
| P3 → P2 → P1 | 79.58 | 97.89 | 96.84 |
| P3 → P2 → P1 with P-FPN | **79.86** | **98.01** | **96.91** |

**Table 4** **Ablation analysis of weight fusion and cross refinement block.** Bold number is the best result.

| Weight fusion | Cross refinement block | F1 (%) |
|---|---|---|
|  |  | 77.82 |
|  | ✓ | 79.58 |
| ✓ |  | 78.62 |
| ✓ | ✓ | **79.94** |

## Ablation experiments between weight fusion and cross refinement block

An ablation experiment between weight fusion and cross refinement block is conducted to further test the effectiveness of the proposed model. Table 4 presents the ablation experimental results. From this table, it can be seen that if only using the weight fusion module, the F1 score can be improved from 77.82% to 78.62%. Similarly, if only using the cross refinement block, a significant improvement is obtained with the F1 score from 77.82% to 79.58%. When two components are synergistically used in our experiments, the model achieves the highest F1 score, reaching 79.94%, which demonstrates the effectiveness of the proposed modules for lane detection.

## Analysis of training loss

The comparative experiments of the training loss are conducted between our proposed method and the CLRNet method (*Zheng et al., 2022*) since the two methods have the similar backbone network. Therefore, the CLRNet method is selected as the baseline method for comparing experiments. The loss curves are plotted between our method and the baseline method during training process are shown in Fig. 4. From the figure, it can be seen that our training loss is lower than the baseline method on the CULane and TuSimple datasets, which can conclude that our improvement strategy is effective.

## Comparison with existing lane detection methods

This study conducted a comprehensive comparative experiments with the existing lane detection methods on the two lane detection datasets, the TuSimple dataset and the CULane dataset.

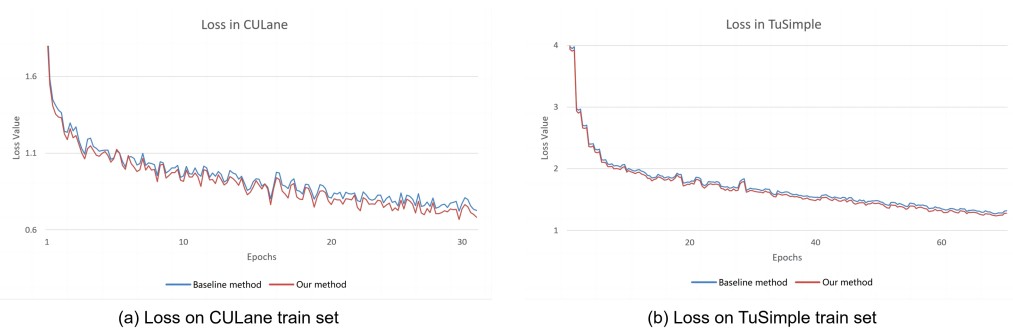

(a) Loss on CULane train set                                    (b) Loss on TuSimple train set

**Figure 4 The result of the training loss between our method and the baseline method.** (A) Training loss on CULane dataset. (B) Training loss on TuSimple dataset.

### Performance on TuSimple

The performance comparison with the current popular methods on the TuSimple dataset is shown in Table 5. From the table, it can be seen that, in all comparative methods, their results are all very good (high value) and their performance difference is very small on this dataset, which shows the result in this dataset seems to be saturated already. However, our method still performs best in the all comparative methods, especially on the Resnet18 backbone, which obtains 98.01% F1 score and 96.91% accuracy. This improvement shows that our lane detection method is effective. In addition, from the Table 5, it also can be seen that, on the Resnet101 backbone, our method obtains a highest false negative (FN) ratio with 3.09%, which shows that our model successfully reduces the learning samples from the wrong lanes and can more accurately determine the wrong lane lines in the detecting process. Comprehensive comparative experiments show that our method outperforms the previous state-of the-art methods in the TuSimple dataset.

### Performance on CULane

The comparative experiments between our method with other popular lane detection methods are conducted in three different backbone networks on the CUlane dataset. As shown in Table 6, using the Resnet18 backbone, our proposed method achieves a new state-of-the-art with an 79.86% F1 score and in most challenging detection scenarios, our method is also significantly superior to other methods. In the meantime, our method can achieve the detecting speed with 157 FPS, which is a competitive detecting speed and is efficient for real-time lane detection.

When using the Resnet34 backbone, the comparative experimental results are shown in Table 7. From this table, it can be seen that our proposed method achieves the highest F1 score of 79.94% and the detecting speed with 126 FPS, which show that our method performs better. Also, in most challenging detection scenarios, our method surpasses other popular methods in the Resnet34 backbone network.

When using a deeper feature extraction network Resnet101, the comparative results are shown in Table 8. From this table, it can be seen that our proposed method obtains the highest F1 score of 80.31% and the fastest detecting speed with 67 FPS. Furthermore,

**Table 5 The comparative experimental results on the Tusimple dataset.** Bold numbers are the best results.

| Method | Backbone | F1 (%) | Accuracy (%) | Proportion of FP (%) | Proportion of FN (%) |
|---|---|---|---|---|---|
| SCNN (*Pan et al., 2018*) | VGG16 | 95.57 | 96.53 | 6.17 | 1.8 |
| RESA (*Zheng et al., 2021*) | Resnet18 | 96.93 | 96.84 | 3.63 | 2.48 |
| LaneATT (*Tabelini et al., 2021a*) | Resnet18 | 96.71 | 95.57 | 3.56 | 3.01 |
| LaneATT (*Tabelini et al., 2021a*) | Resnet34 | 96.77 | 95.63 | 3.53 | 2.92 |
| LaneATT (*Tabelini et al., 2021a*) | Resnet122 | 96.06 | 96.10 | 5.64 | 2.17 |
| PolyLaneNet (*Tabelini et al., 2021b*) | EfficientNetB0 | 90.02 | 93.36 | 9.42 | 9.33 |
| LSTR (*Liu et al., 2021b*) | Resnet18 | — | 96.18 | 2.91 | 3.38 |
| CondLane (*Liu et al., 2021a*) | Resnet18 | 97.01 | 95.48 | 2.18 | 3.80 |
| CondLane (*Liu et al., 2021a*) | Resnet34 | 96.98 | 95.37 | 2.20 | 3.82 |
| CondLane (*Liu et al., 2021a*) | Resnet101 | 97.24 | 96.54 | **2.01** | 3.50 |
| CLRNet (*Zheng et al., 2022*) | Resnet18 | 97.89 | 96.84 | 2.28 | 1.92 |
| CLRNet (*Zheng et al., 2022*) | Resnet34 | 97.82 | 96.87 | 2.27 | 2.08 |
| CLRNet (*Zheng et al., 2022*) | Resnet101 | 97.62 | 96.83 | 2.37 | 2.38 |
| Our method | Resnet18 | **98.01** | **96.91** | 2.31 | 2.12 |
| Our method | Resnet34 | 97.89 | 96.93 | 2.52 | 2.41 |
| Our method | Resnet101 | 97.68 | 96.89 | 2.97 | **3.09** |

in all nine challenging scenarios of the CUlane dataset, our method achieves the best performance, which indicates that our method is easier to reduce the confusion caused by the challenging noise than the state-of-the-art methods when using a deep feature extraction network.

To further demonstrate the computational efficiency of the proposed method, the metric of Floating-Point Operations (FLOPs) is adopted in the experiments. The experimental results are shown in Table 9. From this table, it can be seen that the proposed method obtains a relatively better computational efficiency comparing with the other methods when using Resnet34 backbone. Therefore, the experimental results validate the proposed method is competitive in terms of computational efficiency.

In summary, on CULane dataset, the comparative experiments are conducted in different backbone networks for the lane detection. By analyzing the results from using the backbone networks with Resnet18, Resnet34, and Resnet101, our method obtains the highest F1 score compared to the other popular lane detection methods and also achieves a competitive detection speeds and computational efficiency. Comprehensive analysis shows that our proposed method performs better than the state-of-the-art lane detection methods, which proves the effectiveness and robustness of our proposed lane detection method.

## Analysis of visualization

Samples with challenging weather conditions (such as night with streetlights, strong light reflection, night without streetlights) and good weather condition are selected to visualize and analyze the models using the RESA (*Zheng et al., 2021*), LaneAtt (*Tabelini et al., 2021a*), CondLane (*Liu et al., 2021a*), CLRNet (*Zheng et al., 2022*), and our method on the CULane test set.

Hui et al. (2024), *PeerJ Comput. Sci.*, DOI 10.7717/peerj-cs.1824

**Table 6  Comparative experimental results adopting ResNet18 as the backbone on CULane dataset.** Bold numbers are the best results.

| Method | Backbone | F1 (%) | FPS | Normal (%) | Crowd (%) | Dazzle (%) | Shadow (%) | No line (%) | Arrow (%) | Curve (%) | Cross | Night (%) |
|---|---|---|---|---|---|---|---|---|---|---|---|---|
| SCNN (*Pan et al., 2018*) | VGG16 | 71.60 | 7.5 | 90.60 | 69.70 | 58.50 | 66.90 | 43.40 | 84.10 | 64.40 | 1,990 | 66.10 |
| LaneAF (*Abualsaud et al., 2021*) | ERFNet | 75.63 | 24 | 91.10 | 73.32 | 69.71 | 75.81 | 50.62 | 86.86 | 65.02 | 1,844 | 70.90 |
| LaneAF (*Abualsaud et al., 2021*) | DLA-34 | 77.41 | 20 | 91.80 | 75.61 | 71.78 | 79.12 | 51.38 | 86.88 | 72.70 | 1,360 | 73.03 |
| FOLOLane (*Qu et al., 2021*) | ERFNet | 78.80 | 40 | 92.70 | 77.80 | 75.20 | 79.30 | 52.10 | 89.00 | 69.40 | 1,569 | 74.50 |
| GANet-S (*Morley et al., 2001*) | ResNet18 | 78.79 | 153 | 93.24 | 77.16 | 71.24 | 77.88 | 53.59 | 89.62 | 75.92 | 1,240 | 72.75 |
| LaneFormer (*Han et al., 2022*) | Resnet18 | 71.71 | – | 88.60 | 69.02 | 64.07 | 65.02 | 45.00 | 81.55 | 60.46 | **25** | 64.76 |
| BézierLaneNet (*Feng et al., 2023*) | Resnet18 | 73.67 | 213 | 90.22 | 71.55 | 62.49 | 70.91 | 45.30 | 84.09 | 58.98 | 996 | 68.70 |
| LaneATT (*Tabelini et al., 2021a*) | Resnet18 | 75.13 | **250** | 91.17 | 72.71 | 65.82 | 68.03 | 49.13 | 87.82 | 63.75 | 1,020 | 68.58 |
| CondLane (*Liu et al., 2021a*) | Resnet18 | 78.14 | 173 | 92.87 | 75.79 | 70.72 | 80.01 | 52.39 | 89.37 | 72.40 | 1,364 | 73.23 |
| CLRNet (*Zheng et al., 2022*) | Resnet18 | 79.58 | 119 | 93.30 | 78.33 | 73.71 | 79.66 | 53.14 | 90.25 | 71.56 | 1,321 | **75.11** |
| Our method | Resnet18 | **79.86** | 157 | **93.53** | **78.65** | **74.09** | **80.96** | **53.67** | **90.27** | **72.68** | 1,225 | 74.93 |

Hui et al. (2024), *PeerJ Comput. Sci.*, DOI 10.7717/peerj-cs.1824

**Table 7  Comparative experimental results adopting ResNet34 as the backbone on CULane dataset.** Bold numbers are the best results.

| Method | Backbone | F1 (%) | FPS | Normal (%) | Crowd (%) | Dazzle (%) | Shadow (%) | No line (%) | Arrow (%) | Curve (%) | Cross | Night (%) |
|---|---|---|---|---|---|---|---|---|---|---|---|---|
| SCNN (*Pan et al., 2018*) | VGG16 | 71.60 | 7.5 | 90.60 | 69.70 | 58.50 | 66.90 | 43.40 | 84.10 | 64.40 | 1990 | 66.10 |
| LaneAF (*Abualsaud et al., 2021*) | ERFNet | 75.63 | 24 | 91.10 | 73.32 | 69.71 | 75.81 | 50.62 | 86.86 | 65.02 | 1844 | 70.90 |
| LaneAF (*Abualsaud et al., 2021*) | DLA-34 | 77.41 | 20 | 91.80 | 75.61 | 71.78 | 79.12 | 51.38 | 86.88 | 72.70 | 1360 | 73.03 |
| FOLOLane (*Qu et al., 2021*) | ERFNet | 78.80 | 40 | 92.70 | 77.80 | 75.20 | 79.30 | 52.10 | 89.00 | 69.40 | 1569 | 74.50 |
| RESA (*Zheng et al., 2021*) | Resnet34 | 74.50 | 45.5 | 91.90 | 72.40 | 66.50 | 72.00 | 46.30 | 88.10 | 68.60 | 1896 | 69.80 |
| GANet-m (*Morley et al., 2001*) | Resnet34 | 79.39 | 127 | **93.73** | 77.92 | 71.64 | 79.49 | 52.63 | 90.37 | 76.32 | 1368 | 73.67 |
| LaneFormer (*Han et al., 2022*) | Resnet34 | 74.70 | – | 90.74 | 72.31 | 69.12 | 71.57 | 47.37 | 85.07 | 65.90 | **26** | 67.77 |
| LaneATT (*Tabelini et al., 2021a*) | Resnet34 | 76.68 | **171** | 92.14 | 75.03 | 66.47 | 78.15 | 49.39 | 88.38 | 67.72 | 1330 | 70.72 |
| CondLane (*Liu et al., 2021a*) | Resnet34 | 78.74 | 128 | 93.38 | 77.14 | 71.17 | 79.93 | 51.85 | 89.89 | 73.88 | 1387 | 73.92 |
| CLRNet (*Zheng et al., 2022*) | Resnet34 | 79.73 | 103 | 93.49 | 78.06 | 74.57 | 79.92 | 54.01 | 90.59 | 72.77 | 1216 | **75.02** |
| Our method | Resnet34 | **79.94** | 126 | 93.70 | **78.24** | **74.81** | **81.21** | **54.21** | **90.74** | **73.92** | 1160 | 74.85 |

Hui et al. (2024), *PeerJ Comput. Sci.*, DOI 10.7717/peerj-cs.1824

**Table 8  Comparative experimental results adopting ResNet101 as the backbone on the CULane dataset.** Bold numbers are the best results.

| Method | Backbone | F1 (%) | FPS | Normal (%) | Crowd (%) | Dazzle (%) | Shadow (%) | No line (%) | Arrow (%) | Curve (%) | Cross | Night (%) |
|---|---|---|---|---|---|---|---|---|---|---|---|---|
| SCNN (*Pan et al., 2018*) | VGG16 | 71.60 | 7.5 | 90.60 | 69.70 | 58.50 | 66.90 | 43.40 | 84.10 | 64.40 | 1990 | 66.10 |
| LaneAF (*Abualsaud et al., 2021*) | ERFNet | 75.63 | 24 | 91.10 | 73.32 | 69.71 | 75.81 | 50.62 | 86.86 | 65.02 | 1844 | 70.90 |
| LaneAF (*Abualsaud et al., 2021*) | DLA-34 | 77.41 | 20 | 91.80 | 75.61 | 71.78 | 79.12 | 51.38 | 86.88 | 72.70 | 1360 | 73.03 |
| FOLOLane (*Qu et al., 2021*) | ERFNet | 78.80 | 40 | 92.70 | 77.80 | 75.20 | 79.30 | 52.10 | 89.00 | 69.40 | 1569 | 74.50 |
| LaneATT (*Tabelini et al., 2021a*) | Resnet122 | 77.02 | 26 | 91.74 | 76.16 | 69.47 | 76.31 | 50.46 | 86.29 | 64.05 | 1264 | 70.81 |
| GANet-L (*Morley et al., 2001*) | Resnet101 | 79.63 | 63 | 93.67 | 78.66 | 71.82 | 78.32 | 53.38 | 89.86 | 77.37 | 1352 | 73.85 |
| CondLane (*Liu et al., 2021a*) | Resnet101 | 79.48 | 47 | 93.47 | 77.44 | 70.93 | 80.91 | 54.13 | 90.16 | 75.21 | 1201 | 74.80 |
| CLRNet (*Zheng et al., 2022*) | Resnet101 | 80.13 | 46 | 93.85 | 78.78 | 72.49 | 82.33 | 54.50 | 89.79 | 75.57 | 1262 | 75.51 |
| Our method | Resnet101 | **80.31** | **67** | **94.04** | **78.91** | **74.64** | **82.56** | **54.69** | 89.84 | 75.83 | **1179** | **75.43** |

**Table 9  Comparative experimental results of floating point operations on CULane dataset.** Bold numbers are the best results.

| Method | Backbone | GFLOPs |
|---|---|---|
| SCNN (_Pan et al., 2018_) | VGG16 | 328.4 |
| LaneAF (_Abualsaud et al., 2021_) | ERFNet | 22.2 |
| LaneAF (_Abualsaud et al., 2021_) | DLA-34 | 23.6 |
| RESA (_Zheng et al., 2021_) | Resnet34 | 41.0 |
| LaneATT (_Tabelini et al., 2021a_) | Resnet34 | **18.0** |
| CondLane (_Liu et al., 2021a_) | Resnet34 | 19.6 |
| Our method | Resnet34 | 21.5 |

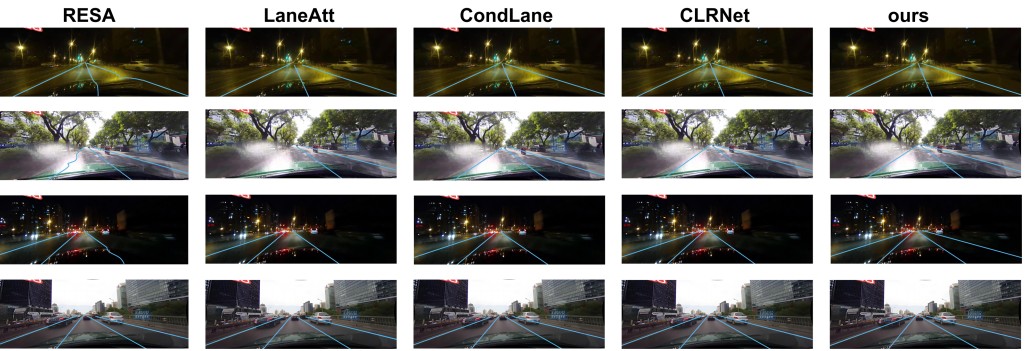

**Figure 5  Visualization result of the RESA, LaneAtt, CondLane, CLRNet, and our method.** Image source: CULane dataset, https://xingangpan.github.io/projects/CULane.html.

As shown in Fig. 5, under the detection condition in night with streetlights in the first row of Fig. 5, it can be observed that the RESA method exhibits an uneven lane detection, whereas the other four methods demonstrate superior detection results. From the second row of Fig. 5, the detection condition is under strong light reflection, it can be seen that the RESA method still suffers from detection distortion and the LaneAtt method fails to detect the left lane line. Whereas our method obtains a good detecting performance. Under the night without streetlights environment, the detecting results are shown in the third row of Fig. 5, it can be observed that our method accurately detected and located the rightest lane line, but the other four methods failed to detect it. In the last row of the figure, it shows that the weather condition is good. Under this condition, all five methods successfully detected the four lane lines, however, our method still exhibits more robust and accuracy. From the visualization analysis, it can be concluded that our method performs better regardless of extreme challenging weather or good weather conditions.

## CONCLUSION

In this article, a novel lane detection method based on Proportional Feature Pyramid Network(P-FPN) is proposed through fusing the weights into the FPN. In the P-FPN network, the cross refinement block and loss block are introduced. The cross refinement

block makes more attention to the feature layers with more knowledge and refines the lanes. The loss block obtains a good performance by regressing the lane as a whole unit. In our method, the high-level features are explored to predict lanes coarsely while local-detailed features are leveraged to improve localization accuracy. Extensive experiments on two widely used lane detection datasets demonstrate the proposed method achieves competitive performance with F1 score of 80. 31% on the CULane dataset and F1 score of 98. 01% on the TuSimple dataset. Comparing with several state-of-the-art approaches, the proposed method performs better in either detection accuracy or efficiency. From the results of the visualization, the proposed method provides more precise lane detection than the other methods.

In future, the proposed method will be further investigated and refined by exploring the more suitable model architectures and balancing the computing time and accuracy.

### Funding
This work was supported by the Project of Educational Commission of Guangdong Province (Grant 312 2021ZDZX1090, 2023ZDZX1082), the China University Innovation Fund (Grant 2021ITA02023), the Shenzhen Polytechnic Project (Grant 6021310017K, 6022310032K, 6021210065K), the Opening Foundation of State Key Laboatory of Cognitive Intelligence, iFLYTEK (Grant COGOS-2023HE07), and the Science and Technology Innovation Project of University of Science and Technology Liaoning (Grant LKDYC202219). The funders had no role in study design, data collection and analysis, decision to publish, or preparation of the manuscript.

### Grant Disclosures
The following grant information was disclosed by the authors:
The Project of Educational Commission of Guangdong Province: 312 2021ZDZX1090, 2023ZDZX1082.
China University Innovation Fund: 2021ITA02023.
Shenzhen Polytechnic Project: 6021310017K, 6022310032K, 6021210065K.
Opening Foundation of State Key Laboatory of Cognitive Intelligence, iFLYTEK: COGOS-2023HE07.
Science and Technology Innovation Project of University of Science and Technology Liaoning:  LKDYC202219.

### Competing Interests
The authors declare there are no competing interests.

### Author Contributions

- Jiapeng Hui conceived and designed the experiments, performed the experiments, analyzed the data, performed the computation work, prepared figures and/or tables, authored or reviewed drafts of the article, and approved the final draft.

- Guoyun Lian conceived and designed the experiments, analyzed the data, authored or reviewed drafts of the article, and approved the final draft.
- Jiansheng Wu conceived and designed the experiments, analyzed the data, authored or reviewed drafts of the article, and approved the final draft.
- Shuting Ge conceived and designed the experiments, analyzed the data, authored or reviewed drafts of the article, and approved the final draft.
- Jinfeng Yang conceived and designed the experiments, analyzed the data, authored or reviewed drafts of the article, and approved the final draft.

## Data Availability

The code for this work is available at Zenodo: Hui, J. (2023). Proportional feature pyramid network based on weight fusion for lane detection. Zenodo. https://doi.org/10.5281/zenodo.10210753

The CULane dateset is available at: https://xingangpan.github.io/projects/CULane.html

The TuSimple dateset is available at:

https://www.kaggle.com/datasets/manideep1108/tusimple?resource=download, Apache License 2.0, https://github.com/TuSimple/tusimple-benchmark/blob/master/LICENSE.

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
