# Peer review of "Proportional feature pyramid network based on weight fusion for lane detection"

_PeerJ Computer Science, doi:10.7717/peerj-cs.1824_

## Round 0.1 · original submission · Minor Revisions

Dear authors,

Your paper has been reviewed by two reviewers who asked for revision of your paper. Please correct the paper and provide a cover letter with replies point to point.

Reviewer 1 has suggested that you cite specific references. You are welcome to add it/them if you believe they are relevant. However, you are not required to include these citations, and if you do not include them, this will not influence my decision.

Reviewer 1 ·

Basic reporting

It is a very interesting and significantly research, based on an application of artificial intelligence in the detection of traffic lanes... The paper represents an extremely important segment of research in road engineering. The abstract of the paper describes the problem extremely well and indicates the objectives. The authors present an excellent and contemporary literature review.

Experimental design

Special praise to the author for the methodology of the paper and the given case study. To make the work even better, I give certain suggestions:

Keywords - Change keywords in the paper. A few words are in the title of the paper

In the abstract of the paper, avoid abbreviations and write full name (Line 23 – CULane & TuSimple dataset).

Figure 1. - I am of the opinion that Figure 1 should be placed below Line 58.

Figure 2. - I am of the opinion that Figure 2 should be placed below Line 112.

I suggest the authors to change the RELATED WORKS chapter to Literature review. Also, transfer some significant cited literature from the Introduction to this chapter.

At the end of the introduction, it is necessary to explain the rest of the paper by chapter (2-3 sentences).

Validity of the findings

I suggest the authors to include the following literary source in the Literature review:

Jayapal, P. K., Muvva, V. R., & Desanamukula, V. S. (2023). Stacked Extreme Learning Machine with Horse Herd Optimization: A Methodology for Traffic Sign Recognition in Advanced Driver Assistance Systems. Mechatron. Intell Transp. Syst., 2(3), 131 145. https://doi.org/10.56578/mits020302

M. Subotić, E. Softić, V. Radičević, and A. Bonić, “Modeling of operating speeds as a function of longitudinal gradient in local conditions on two-lane roads,” Mechatron. Intell Transp. Syst., vol. 1, no. 1, pp. 24-34, 2022. https://doi.org/10.56578/mits010104.

I suggest the authors to add 2-3 sentences about the comparison of the obtained results in the conclusion. Without that part, the conclusion is not enough.

In some parts of the paper, the first person plural is used to write the paper (Line 71, 74, 81, 139, 142, 157, 177, 178, 211, 214, 215, 223,224,226,233, 245, 265,292, 307). That needs to be corrected. The paper can not be written in the first person!

Additional comments

No comment

Cite this review as

Reviewer 2 ·

Basic reporting

The article "Proportional Feature Pyramid Network based on Weight Fusion for Lane Detection" addresses the challenges of lane detection under extreme conditions. The proposed Proportional Feature Pyramid Network (P-FPN) introduces a novel approach to address issues associated with existing methods such as large model sizes, embedding difficulties, and slow detection speeds. The key innovation lies in the fusion of weights into the Feature Pyramid Network (FPN) for lane detection.

Experimental design

One notable aspect of the P-FPN is the incorporation of a cross refinement block, which enhances the accuracy of lane detection. This block takes feature maps and anchors as inputs and progressively refines the anchors from high to low-level feature maps. This approach ensures that high-level features contribute to coarse lane predictions, while local-detailed features refine the localization accuracy. This strategy allows the model to capture crucial pixels and predict complex lane line topologies effectively.

Validity of the findings

The experimental evaluation on the CULane and TuSimple datasets is a strength of the article. The comparison with state-of-the-art approaches demonstrates the competitiveness of the proposed method. The emphasis on widely used datasets enhances the generalizability of the findings, reinforcing the credibility of the proposed P-FPN.

Additional comments

However, the article could benefit from a more detailed discussion on the computational efficiency of the proposed method. While the paper mentions the issues of model size and detection speed in existing methods, it would be valuable to provide specific metrics and comparisons to highlight the improvements achieved by the P-FPN.

Cite this review as

---

## Round 0.2 · Minor Revisions

Dear authors,

Your revised version of the paper has been reviewed by two experts who take part in reviewing of original version of the paper. One of them, reviewer 2 asked for additional minor revision. Please revise your paper again, mark corrections, and provide a cover letter with replies to the reviewer.

Reviewer 1 ·

Basic reporting

I accept all corrections

Experimental design

I accept all corrections

Validity of the findings

I accept all corrections

Additional comments

I accept all corrections

Cite this review as

Reviewer 2 ·

Basic reporting

The paper provides a clear and concise overview of the challenges in lane detection under extreme conditions and introduces the Proportional Feature Pyramid Network (P-FPN) as a solution. The abstract effectively communicates the motivation, methodology, and key results. The language is accessible to readers with a background in the field, making it suitable for a wider audience.

The inclusion of relevant datasets, Chinese Urban Scene Benchmark for Lane Detection (CULane) and TuSimple Lane Detection Challenge (TuSimple), enhances the paper's credibility. However, it would be beneficial to provide more details on the datasets, such as size, characteristics, and any preprocessing applied.

Experimental design

The paper presents a novel approach with the introduction of the P-FPN and the cross refinement block, demonstrating a thoughtful design to address challenges associated with existing methods. The utilization of weight fusion into the FPN for lane detection is a notable innovation.

The methodology is adequately explained, especially the integration of the cross refinement block, which refines anchors progressively. However, more details on the hyperparameters and training strategies would contribute to the reproducibility of the results. A comprehensive explanation of the choice of evaluation metrics would also enhance the experimental design section.

Validity of the findings

The experimental results on CULane and TuSimple datasets illustrate the effectiveness of the proposed method, showcasing competitive performance against state-of-the-art approaches. The findings align with the paper's objectives and contribute to the advancement of lane detection techniques.

However, to strengthen the validity of the findings, additional ablation studies or sensitivity analyses could be conducted to assess the impact of individual components introduced in the P-FPN. This would help in understanding the specific contributions of weight fusion and the cross refinement block to the overall performance.

Cite this review as

---

## Round 0.3 · accepted · Accept

Dear authors,

Your paper revised through two rounds of reviews can be accepted because you have addressed all of the reviewers' comments.